

# Some fundamental elements for studying social-ecological co-existence in forest common pool resources

Jean-Baptiste Pichancourt

Université Clermont-Auvergne, INRAE, Complex Systems Lab (UR LISC), Centre de Clermont-Ferrand, Aubière, France

Corresponding author
Jean-Baptiste Pichancourt,
jean-baptiste.pichancourt@inrae.fr

## ABSTRACT

For millennia, societies have tried to find ways to sustain people's livelihoods by setting rules to equitably and sustainably access, harvest and manage common pools of resources (CPR) that are productive and rich in species. But what are the elements that explain historical successes and failures? Elinor Ostrom suggested that it depends on at least eight axiomatic principles of good governance, whereas empirical results suggest that these principles are not sufficient to describe them, especially when applied to CPRs that possess great social and ecological diversity. The aim of this article is to explore the behavior of a mathematical model of multi-species forest dynamics that respects the foundations of ecology and Ostrom's governance theory, in order to detect possible constraints inherent to the functioning of these complex systems. The model reveals that fundamental structural laws of compatibilities between species life-history traits are indeed constraining the level of co-existence (average and variance) between a diversity of co-vulnerable timber resource users (RU) and of competing tree species. These structural constraints can also lead to unexpected outcomes. For instance in wetter forest commons, opening up the access to as many diverse RUs as there are competing tree species, produces a diversity of independently-controlled disturbances on species, collectively improving the chances of coexistence between species with different life-history traits. Similar benefits are observed on forest carbon and on profits from timber harvesting. However in drier forest commons, the same benefits cannot be observed, as predicted on the basis of the constraining laws. The results show that the successes and failures of certain management strategies can be reasonably explained by simple mechanistic theories from ecology and the social-ecological sciences, which are themselves constrained by fundamental ecological invariants. If corroborated, the results could be used, in conjunction with Ostrom's CPR theory, to understand and solve various human-nature coexistence dilemmas in complex social-ecological systems.

## INTRODUCTION

### Socio-ecological context

An estimated 350 million, including 60 million indigenous people, live in or adjacent to forests, and are almost wholly dependent on forest biological resources for their subsistence and income needs. Among these people, many manage a diversity of tree species, *e.g.*, for timber and firewood, as "*common-pool resources*", or CPR for short (*Ostrom, 1990*; *Agrawal & Gibson, 1999*; *Toledo et al., 2003*; *Chhatre & Agrawal, 2008*; *Persha, Agrawal & Chhatre, 2011*; *Kimengsi et al., 2019*). CPR institutional arrangements were established to ensure that a greater diversity of resource users can equitably and sustainably access, extract and share a diversity of resource species, but under some collectively accepted rules (*Ostrom, 1990*; *Agrawal & Gibson, 1999*; *Kijima, Sakurai & Otsuka, 2000*; *Klosowski et al., 2001*; *Lemos & Agrawal, 2006*; *Antinori & Rausser, 2007*; *Chhatre & Agrawal, 2008*; *Parrotta & Agnoletti, 2012*), and properly manage the diversity of social traits and cultural practices between resource users (*Poteete & Ostrom, 2004*; *Naidu, 2009*; *Saunders, 2014*; *Betts et al., 2021*). However, CPR forests that are rich in co-vulnerable resource users and species are complex social-ecological systems (SES), and their functioning is difficult to understand and control (*Poteete & Ostrom, 2004*; *Naidu, 2009*; *Saunders, 2014*; *Betts et al., 2021*). It is therefore no surprise to observe that not all CPR institutional arrangements can achieve their socio-economic objectives, and even less so when they are combined with other ecological objectives such as those related to the protection of biodiversity and the sequestration of carbon for the purpose of mitigating climate change. Some institutional CPR principles are known to be useful predictors of the successes and failures at solving these complex human-nature CPR dilemmas (*Ostrom, 1990*; *Ellis & Allison, 2004*; *Poteete & Ostrom, 2004*; *Chhatre & Agrawal, 2008*; *Naidu, 2009*; *Persha, Agrawal & Chhatre, 2011*; *Saunders, 2014*; *Betts et al., 2021*). Nevertheless, even when all of them are respected, there are still unexplained failures (*Ellis & Allison, 2004*; *Cox, Arnold & Tomás, 2010*; *Saunders, 2014*).

### Scientific problem

Understanding the causal factors would require experimental protocols and world-scale evidence, both of which are rarely available. This is due to the long time frame and high cost required to monitor processes necessary to produce correlations between a wide range of tree species diversity and of CPR access rates for a broad diversity of co-vulnerable resource users, themselves collectively characterized by a similar diversity of individual management practices. Under these conditions, theoretical modeling is essential to generate a set of simple and falsifiable predictions. However, the dimensions of such models can be so large and the outcomes so sensitive to many social-ecological drivers, that it is tempting to first search for guiding principles. For instance, it is assumed that the functioning of such complex SES are constrained by internal structural invariants, independent of many external factors, and that if properly described, they should simplify our understanding of the relative success and failure of simultaneously achieving the multiple above-mentioned objectives.

## Aim and structure of the article

The aim of this article is to explore the behavior of a mathematical model of multi-species forest dynamics that respects the foundations of ecology and Ostrom's governance theory, in order to detect possible constraints inherent to the functioning of these complex systems. To achieve this, it is shown that we benefit from re-framing the general social-ecological problem (*i.e., the constraints on the social-ecological sustainability of biodiverse productive CPRs accessed by a diversity of resource users*) into a new ecological hypothesis that is far easier to model and obtain knowledge from, *i.e.,* "*how the diversification of (management) disturbances imposed on a pool of competing tree species (by a diversity of RUs) is predicted to impact their coexistence, and functional relationship with the production of a diversity of ecosystem services.*" Based on this re-framing, an existing mathematical model of multi-species forest dynamics (*Pichancourt et al., 2014*) is strategically extended according to the design principles of the Coupled Infrastructure System (CIS) modeling framework (*Anderies, Janssen & Ostrom, 2004*; *Anderies, Barreteau & Brady, 2019*; *Bernstein et al., 2019*). The latter framework is used to control the effect of Ostrom's principles of good governance of CPRs, and to limit the analysis more to the ecological processes than to the social processes. Numerical simulations predict a set of phenomenological relationships between various incomes and outcomes: the forest access rate, the level of diversification of human-controlled disturbances (harvesting, replanting), various forest outcomes (carbon biomass, the Shannon index of tree functional trait diversity, benefits from timber and firewood, *etc.*), and the average functional/life-history traits measured at the tree-assemblage level. Some of these relationships are then discussed in light of widely used ecological theories, in relation to their role as structural constraints on the functioning and viability of these complex SES. Finally, a research agenda is presented for using these new theoretical elements to understand and solve a set of coexistence dilemmas in SES.

# METHODS

## General structure of the model

The structure of the mathematical model follows the guidelines specified by the Coupled Infrastructure System (CIS) framework (*Anderies, Janssen & Ostrom, 2004*; *Anderies, Barreteau & Brady, 2019*; *Bernstein et al., 2019*). The qualitative CIS representation of our problem is summarized in Fig. 1.

The CIS framework is used to model the most complex situations of governance, by braking SES down into four interacting infrastructures. However, not all of them are present in all the SES (see *Anderies, Barreteau & Brady, 2019*). In our case, the model uses only three of them: the forest natural resource infrastructure (RI, containing an assemblage of tree species); the resource users (RU, who access the forest to harvest and manage the tree species for their biomass); and the public infrastructure (PI, representing the collective association, material, rules, finance by and for the RU). The model does not define the fourth one, the governance infrastructure (GI), since there is no need in this study to define
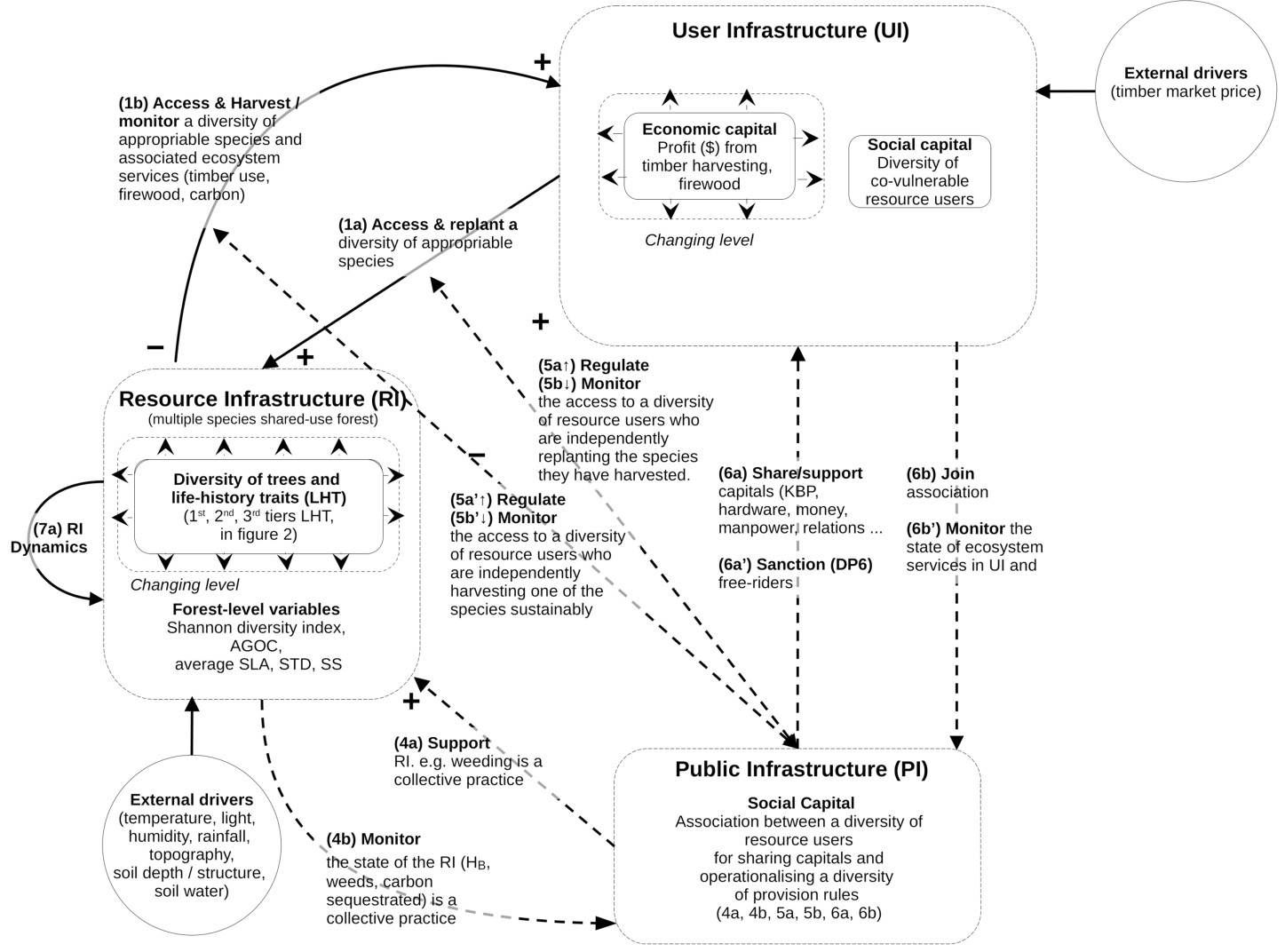

**Figure 1  Conceptual model of coupled infrastructure system (CIS) framework derived from Ostrom's theory on common pool resources.** The model decomposes any social ecological system into various interlinked infrastructures. Here is represented the resource infrastructure (RI), the resource user infrastructure (RU), the public Infrastructure (PI). These four infrastructures are linked through operational and institutional processes specified in the figure (1a to 7a). The model does not define any governance infrastructure (GI), as in this study there is no need to define providers of institutional rules and budget for the PI or RU. For this reason some of the links usually specified in CIS model (links 2a, 2b, 3a and 3b) are not visible in Fig. 1. However, as this coding is considered standard when reading CIS, it was decided that the specific CIS system of this study would use the same numbers of the functional links. The mathematical equations and variables associated with these links and infrastructures is described in the method section.                                         

providers of institutional rules and budgets for the PIs or RUs. This structure presented in Fig. 1 is used throughout the Methods section.

## Model of resource infrastructure (RI) and its internal dynamics (link 0a in Fig. 1)

The dynamics of the forest RI under management and environmental constraints is determined by a mathematical model of multi-species forest ecosystem dynamics. The RI model was developed by *Pichancourt et al. (2014)* and uses the same parameterization here

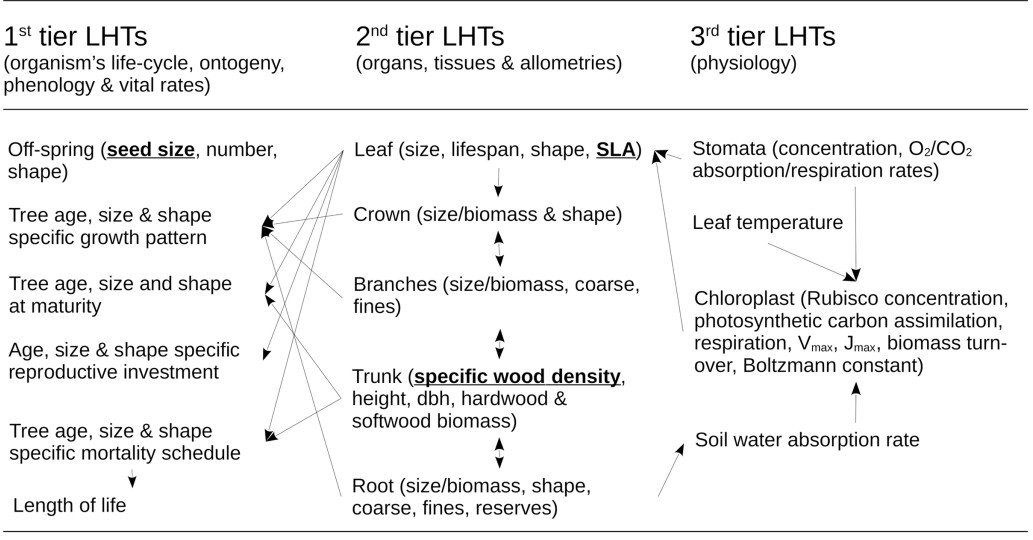

**1st tier LHTs**
(organism's life-cycle, ontogeny, phenology & vital rates)

**2nd tier LHTs**
(organs, tissues & allometries)

**3rd tier LHTs**
(physiology)

Off-spring (**seed size**, number, shape)

Tree age, size & shape specific growth pattern

Tree age, size and shape at maturity

Age, size & shape specific reproductive investment

Tree age, size & shape specific mortality schedule

Length of life

Leaf (size, lifespan, shape, **SLA**)

Crown (size/biomass & shape)

Branches (size/biomass, coarse, fines)

Trunk (**specific wood density**, height, dbh, hardwood & softwood biomass)

Root (size/biomass, shape, coarse, fines, reserves)

Stomata (concentration, $O_2/CO_2$ absorption/respiration rates)

Leaf temperature

Chloroplast (Rubisco concentration, photosynthetic carbon assimilation, respiration, $V_{max}$, $J_{max}$, biomass turn-over, Boltzmann constant)

Soil water absorption rate

**Figure 2 Structure of the ecological model for the resource infrastructure.** Multiple tier functional model of species life history traits (LHTs) used to estimate the average population growth (r: Darwinian fitness) of every tree species (summary from *Pichancourt et al. (2014)*), based on life-history theory.

(see http://onlinelibrary.wiley.com/doi/10.1111/gcb.12345/suppinfo). Its biological structure is detailed in Fig. 2. It includes sub-models of life-history traits, architecture, physiology, demography, competition for resources, and above-ground organic carbon biomass in relation to climate, soil and landscape contexts. This model was used because it is still one of the only mathematical models as of today to be able to correctly scale up the dynamics of single to complex tree species assemblages, and to make predictions on their structure and dynamics under a diversity of forestry or restoration practices. Because its underpinnings are important to interpret the results, this part of the Methods section presents a summary of its structure and basic assumptions.

### Model of life-history traits and population dynamics at the species level

The model of *Pichancourt et al. (2014)* is primarily based on the life-history theory. According to this theory, biophysical constraints on the allocation of energy between reproduction, growth and self-maintenance are viewed as the primary explanation for why species do not possess arbitrary combinations of life-history traits (LHTs) between organs and life-stages, throughout the life cycle of the organism. These ultimately drive the population growth rate of species to adapt to their environmental conditions. To reflect this principle, the model is structured according to a multiple-tier approach to LHTs (Fig. 2).

The first tier of LHTs represents the specific vital rates of stage and size throughout the life cycle of a tree species (*i.e.*, seed survival, germination, tree growth, survival and fertility). These traits are constrained by allometric traits that are assumed to be optimally defined by natural selection (the second tier of LHTs, as outlined by the scaling theory of ecology). Finally, the second tier of LHTs is itself constrained by the third tier—the metabolic LHTs—based on the different physiological processes, *e.g.*, photosynthetic

carbon assimilation, respiration, $V_{max}$, $J_{max}$, biomass turnover, water absorption, carbon biomass production (as outlined, *e.g.*, by the metabolic theory of ecology).

For plant species, the theory also predicts that ~50% of the variability of most of the tree LHTs on Earth can ultimately be reduced to three ((*van Bodegom, Douma & Verheijen, 2014*): specific leaf area (SLA) ($m^2.kg^{-1}$); specific wood density (SWD) ($kg.m^{-3}$); and seed size (SS) (kg). Under this realistic assumption, species with similar values of these three LHTs share other similar 1–2- and 3-tier LHT and life-cycle strategies. Based on this organization, mathematical models can be developed to create a wide range of unique tree species life cycle strategies (see summary of models in *Pichancourt et al. (2014)* and in *van Bodegom, Douma & Verheijen (2014)*). In this article, computational capabilities limited our exploration to eight species, representing all the combinations between a range of extreme values of LHTs found in the literature (see *Pichancourt et al., 2014*): SLA (2.5–20 $m^2.kg^{-1}$); SWD (400–1,000 $kg.m^{-3}$); and SS ($10^{-7}$–$10^{-3}$ kg per seed).

### Model of species competition

This type of model can be mathematically generalized to correctly scale up the dynamics of tree communities within the stand. Vertical competition in relation to light is simply modeled from the principle of perfect plasticity (PPA), used in all modern terrestrial biosphere models. Competition for water in the soil is also simply modeled using a soil water bucket model that depends on soil characteristics. In the present study, the RI model was parameterized like in *Pichancourt et al. (2014)* to match forests where the soil is a 2-m-deep loam A–B horizon composed of 50% sand, 20% clay and 30% silt, which is the most common soil found on Earth.

### Indicator of the forest state

Forest-level variables were then calculated per square meter and re-scaled to account for the forest area considered (1 ha). This study focused on the Shannon index of the diversity of life-history traits (based on the relative density of trees belonging to every combination of LHT values: SLA/SWD/SS), the organic above-ground carbon sequestration of the trees (MgC/ha/y), the live biomass and deadwood in the forest (t/ha/y) and, finally, the average forest level of LHTs (SLA, SWD, SS). Similar to *Pichancourt et al. (2014)*, the model exemplified the projected state of the forest RI under 100-year climate scenarios where sub-tropical forests were parameterized based on the latitude around Brisbane in Queensland, Australia, and were subject to one of the predicted scenarios of hotter and drier climate change (CSIRO mk3.5 using CMIP3 model: see details in *Pichancourt et al. (2014)*: Table S1).

## Model of resource users (RU in Fig. 1)

Under the CIS framework, the model first defines the boundaries of livelihood diversification in RUs by setting the maximal diversity of RUs that can access the RI to harvest and replant tree species. The RI here was intentionally limited to eight species for computational reasons (the species are presented in "Model of resource infrastructure (RI) and its internal dynamics (link 0a in Fig. 1)" of Methods). Therefore, the minimal and maximal number of RUs are zero and eight, respectively. Within these constraints, the

model holds that from zero to eight RUs can access the RI to independently and sustainably harvest (link 1a in Fig. 1) one species per user (*i.e.*, from zero to eight species-user pairings) to sell the biomass on the international market. Each harvesting rate is defined by econometric equations that integrate variables in relation to forest biomass, species life-history (functional) traits, and socio-economic and physical factors (Eqs. (1)–(15) in "Model of resource users (RU in Fig. 1)"). Similarly, RUs impose post-harvesting planting strategies under certain rates (link 1b, Eqs. (16)–(18) in "Model of timber harvesting (link 1a in Fig. 1)"), as well as the harvesting of deadwood for individual heating (link 1a, Eq. 19 in "Model of fuel-wood harvesting (link 1a′ in Fig. 1)").

### *Model of timber harvesting (link 1a in Fig. 1)*

There is a large spectrum of harvesting behaviors in forest systems on Earth. However, in the absence of global data and a model to describe harvesting behavior in various types of forest institutional agreements, the choice was made to use the dataset and model of *Canham, Rogers & Buchholz (2013)*. This model represents the most accomplished effort to date to predict the probability of individual RUs of harvesting trees in forests in the United States. It describes a succession of decisions that emerge at three scales of observation: at the stand level (biomass and basal area), at the species level (21 species with a wide spectrum of LHTs), and at the individual tree level (size). The main interest of this dataset and model is that the harvesting rates are assumed to match every species' ecological state and biological characteristic considered in the RI model. For this reason, any harvesting model combined with the multi-species forest model is assumed to independently apply sustainable harvesting rates on every species, but without making prior assumptions about their impact on the demographic structure and dynamics of other competing species not directly impacted by each specific harvesting rate. Therefore, applying a diversity of species-specific harvesting rates, simultaneously or as a temporal succession, is expected to impact the coexistence of tree species within the forest in a complex way.

Consistent with *Canham, Rogers & Buchholz (2013)*, the harvesting behavior of individual RUs can be modeled as a hierarchical chain of four decisions:

1. Decision 2.3.1.1 (stand level): the RU decides whether or not to harvest given the biomass state of the forest and the average timber density of the trees.
2. Decision 2.3.1.2 (stand level): the RU determines the average basal area harvested given the average timber density of the trees.
3. Decision 2.3.1.3 (species level): the RU determines the probability for every tree species to be harvested based on its life-history traits (SWD, SS/maximal height of the tree species, SLA).
4. Decision 2.3.1.4 (individual tree level): the RU determines the probability for every tree to be harvested given its size (diameter at breast height *dbh*).

Using this approach, it becomes possible to predict the supply of timber for every species at a given year of harvest. These four decisions are now explained in detail.

*Decision 2.3.1.1—stand level preference: what is the probability that the plot is logged?*
If the RU decides to harvest during a lifetime, the decision to harvest depends on the time T since the last decision and the above-ground biomass B of the adult trees in the stand, since the greater the biomass and the longer the time since the last harvest is, the greater the probability $P_{harvest}$ will be. In keeping with *Canham, Rogers & Buchholz (2013)*, this process can be simply modeled as follows:

$$P_{harvest} = 1 - a_1 e^{\left(-m_1 B^{b_1}\right)^T} \text{ with } a_1 = 1 \tag{1}$$

where coefficients $a_1$, $b_1$ and $m_1$ were measured per forest type and $P_{harvest}$ is a function bounded between 0 and 1 that asymptotically reaches 1. In keeping with Cunham, the RU harvests the plot when the forest biomass reaches $P_{harvest} \geq 95\%$.

On the basis of the original dataset of *Canham, Rogers & Buchholz (2013)*, the dominant species mentioned is associated with the forest type. Therefore, using the species-level specific wood density (SWD) reported in the U.S. Forest Service database (http://www.feis-crs.org/beta/), or other sources when not accessible on the site (Table S1), it is possible to estimate the average forest-type level $SWD_{mean}$. By doing so, we can easily predict from the original dataset of *Canham, Rogers & Buchholz (2013)* that the $SWD_{mean}$ of a stand is a good predictor of Canham's coefficients $b_1$ and $m_1$ on the basis of Eq. (1) (Table S1, Eq. (2)) and, consequently, of how the RU made her/his first decision at that level:

$$b_1 = -0.0016 \pm 0.002 SWD_{mean} + 1.61 \pm 1.36 \; (R^2 = 0.84) \tag{2}$$

$$m_1 = 1.21^{-5} \pm 1.11^{-5} SWD_{mean} - 0.0046 \pm 0.007 \; (R^2 = 0.92) \tag{3}$$

*Decision 2.3.1.2—stand level preference: how much basal area is harvested per stand?*
Once the harvesting decision for the forest is made, the RU determines the percentage of basal area removed (RBA), given the above-ground biomass of the stand (or stand basal area):

$$RBA = a_2 e^{\left(-m_2 B^{b_2}\right)} \tag{4}$$

Like in *Canham, Rogers & Buchholz (2013)*, coefficients $a_2$, $b_2$ and $m_2$ are measured per forest type, the same way as in Decision 1 (Table S2). By doing so, it is clear that $b_1$ and $m_1$ can be predicted again from $SWD_{mean}$ (Table S1), such that:

$$b_2 = -0.003 \pm 0.003 SWD_{mean} + 2.44 \pm 2.25 \; (R^2 = 0.87) \tag{5}$$

$$m_2 = 0.0007 \pm 0.002 SWD_{mean} - 0.33 \pm 1.35 \; (R^2 = 0.51) \tag{6}$$

$$a_2 = 114.5 \pm 156 \, m_2 + 40.9 \pm 25.2 \; (R^2 = 0.84) \tag{7}$$

Corroborating the fact that SWD is the main factor for timber pricing at the international level (Fig. S1) and, therefore, the driving force behind the decision to harvest at the stand level (Decisions 1 and 2).

*Decision 2.3.1.3—species-level preference: which species is harvested?*
After the RU determines the basal area to harvest, (s)he chooses the tree species to be harvested. *Canham, Rogers & Buchholz (2013)* predicted that the probability of a given species to be harvested depends on the basal area removed (*RBA*):

$$P_{hs} = 1 - \gamma e^{(-\beta RBA)^{\alpha}} \tag{8}$$

where coefficients $\alpha$, $\beta$ and $\gamma$ are measured at the species level. Using the same original data, combined with general information found on http://www.feis-crs.org/beta/ and from other sources when not accessible on the site (Table S3), it was possible to perform a simple analysis of how RUs made their decisions based on specific LHTs (SWD, SLA, and/or SS that is highly correlated with tree size at maturity; *Pichancourt et al., 2014*). This time, $\alpha$ and $\beta$ from Eq. (8) could also be reasonably well predicted from the species-specific SS, *i.e.*, maximal height of the tree species (Fig. S1), such that:

$$\alpha = 0.12 \pm 0.08 \, logSS + 2.98 \pm 0.85 \, (R^2 = 0.35) \tag{9}$$

$$\beta = 0.5 \pm 0.45 e^{(-3.45 \pm 0.37\alpha)} \, (R^2 = 0.82) \tag{10}$$

$$\gamma = -0.095 \pm 0.03\alpha + 1.1 \pm 0.05 \, (R^2 = 0.74) \tag{11}$$

These relationships reveal that at this step, after SWD, RUs now consider the maximal height of a tree species (since, on average, species with larger seeds grow into taller tree species, and *vice versa*; *Pichancourt et al., 2014*). By doing so, we show that we can simplify the harvesting model on the basis of our knowledge of the tree LHTs, at least when using the data of *Canham, Rogers & Buchholz (2013)*.

*Decision 2.3.1.4—tree-level preference: which size-class is harvested?*
Once the proportion of a species is determined, the RU decides whether each tree should be logged based on the diameter at breast height of the tree (*dbh*), which depends on tree height:

$$P_{ht} = e^{\frac{-1}{2}\left(\frac{dbh - \mu}{\sigma}\right)^2} \quad \text{with} \tag{12}$$

$$\sigma = a_2 + b_2 RBA^c \tag{13}$$

$$c = 1$$

$$\mu = 1.058 \pm 0.77 \, log(SS) + 54.78 \pm 3.68 \, (R^2 = 0.345) \tag{14}$$

**Combination of harvesting decisions**

Once we have the models for the four decisions, we can combine them to predict the probability that any tree of any species *i* is harvested in the forest given the biomass of trees in the forest on the basis of *SWD* and SS values:

$$P_{i,\text{SWD,SS}} = \left(1 - \gamma e^{(-\beta \text{RBA})^\alpha}\right) e^{\frac{-1}{2}\left(\frac{\text{dbh}-\mu}{\sigma}\right)^2} \tag{15}$$

This model predicts the supplied timber biomass for each species at a given year of harvest, excluding the mahogany tree. On the basis of this prediction, we can deduce the expected revenue from timber harvesting ($/ha/y) since the international price for timber is highly based on the SWD value of species (*Ahmed & Ewers, 2012*; Fig. S3). In this case, the model assumes a constant price for this analysis, which can be changed in the model. Furthermore, the model considers that all RUs have the equipment necessary to properly cut down their trees, and that the costs of increasing the number of RUs and thus of species harvested does not increase based on the assumption that the community-level marginal return on labor for each RU represents only a small fraction of the value of the marginal product of labor to manage more species (*Agrawal & Gibson, 1999*; *Kijima, Sakurai & Otsuka, 2000*).

### Model of fuel-wood harvesting (link 1a' in Fig. 1)

Fuelwood remains a crucial resource for rural livelihoods in many countries (*Foley, 1985*; *Webb & Dhakal, 2011*). Following *Foley (1985)* and *Webb & Dhakal (2011)*, the model considers fuelwood harvesting as a constraint on timber harvesting. It assumes that the harvesting of fuelwood begins with deadwood and is then supplemented with live trees if deadwood is not available. At the global scale, fuelwood demand can vary from 200 kg to more than 2,000 kg per person per year. The model defines the main requirements in fuelwood as a function of the size of the household ($HH_{size}$) and the density of households per hectare ($HH_{density}$), but the model can be changed and adapted to the local context:

$$B_{fuelwood} = HH_{density}\, HH_{size}\, (899.24 - HH_{size}\, 34.68) \tag{16}$$

For this study, population density was simply defined as one household per hectare with five members per household.

*Decision 2.3.2.1—preference in species collection*

The fuelwood preferences of the RU has been found to be positively correlated with the fuelwood value index (FVI) of the tree species, which depends on the SWD (*Ramos et al., 2008*), such that:

$$FVI = 0.0475 \pm 0.008\, SWD - 11.424 \pm 4.8 \; (R^2 = 0.82, n = 38) \tag{17}$$

The model assumes a perfect relationship between FVI and harvesting choice, such that the probability of harvesting a given timber species (dead or alive) is considered as the FVI rank of the given species, divided by the sum of all the FVI ranks of all the species present in the forest.

*Decision 2.3.2.2—preference in tree size collection*
Once the RU selects a species to harvest based on the SWD, we can simply predict from the free data of *Thapa & Chapman (2010)* and *Nguyen et al. (2014)* that the collection of deadwood is also based on the tree *dbh*:

$$P_{\text{fwt}} = 12.43 \frac{1}{1.23\sqrt{2\pi}} (100 \, \text{dbh} - 4.106) e^{\left( \frac{-(\log(100 \, \text{dbh} - 4.106) - 3.381)^2}{2 \times 1.23^2} \right)} \quad (R^2 = 0.94) \qquad (18)$$

The combination of these models determines how many trees are harvested. The model then considers harvesting once a year, and that live trees are harvested if the total annual quantity of dead fuelwood for the household (defined in Eq. (16)) is not fulfilled. In all our simulations, this level was never reached since the number of households was low (one per hectare).

### Behavioral model for individual post-harvesting planting (link 1b in Fig. 1)
The model considers that RUs who harvest trees for timber are involved in individual post-harvesting planting. For instance, *Sakurai et al. (2004)* show that the density of trees replanted can be simply modeled on the basis of a few variables: whether the parcel is managed privately, as an association or publicly (PRIVATE: binary 0-1; if 0, then RUs are organized as a community or are part of a public agency for post-harvesting practices); the time it takes to walk to the forest $T_{\text{wt}}$; and soil quality (with sand: $S_{\text{sandy}}$ (binary 0-1); gravel $S_{\text{gravel}}$ (binary 0-1)). Given that the soil considered is loam, and that the time to walk to the forest is 18.5 min, the planting density after harvest $D_{\text{planting}}$ (given as 1,000 seedlings/ha) is defined as follow:

$$D_{planting} = 10.2 \pm 3.32 \, PRIVATE - log(T_{wt}) - 1.85 \pm 0.93 \, S_{gravel} - 2.03 \pm 1.31 \, S_{sandy} \sim 7 \quad (19)$$

The variable PRIVATE = 1, given that it is assumed that the RUs individually replant according to their harvesting preferences defined in Sections B.2 and B.3.

## Model of public infrastructure (PI in Fig. 1)
Every species-user pairing in the model is assumed to be independently sustainable in terms of harvesting and replanting rates. All these activities are also assumed to be perfectly performed on each species, without directly damaging other species (*e.g.*, through blind logging, infrastructure building or transportation). However, the harvesting and replanting disturbances by every species-user pairing are performed independently of the indirect consequences on other species-user pairings. Therefore, given that coexisting species (managed or not) compete for the same physical resources (light, soil, water), then all the species-user pairings can be considered as co-vulnerable and, consequently, the collective economic outcomes from harvesting, tree forest biodiversity and carbon outcomes.

Under these conditions, empirical observations show that co-vulnerable RUs usually form associations and cooperate (link 6b in Fig. 1) to establish common rules of good

conduct and to share useful capitals (knowledge, money, equipment, people) in order to avoid the tragedy of the commons (*Cox, Arnold & Tomás, 2010*). These public infrastructures (PI) are encoded into the CIS as collective practices: link 4a (Fig. 1) represents the collective monitoring by RUs of the ecological state of RI (levels of tree biodiversity, organic carbon biomass), and link 6a (Fig. 1) represents the same monitoring effort but on the state of harvested resources supplied to RUs (effective firewood supply, timber supply and revenues). This PI also assumes that the association of RUs can monitor and control RI access by a collectively accepted quantity of RUs, and thus avoid free-riding (monitor: link 5a, control: link 5b). The PI model assumes that RUs collectively support the RI state through the control of invasive species (link 4b, Eqs. (20)–(23)). In order to focus on the impact of the environmental context, this first article considers that all the RUs form a perfect association (probability P(6a) = 1 (on a 0-1 scale)), leading to the best possible monitoring (P(4a) = 1, P(5a) = 1) and control (P(4b)→max) efforts (Eqs. (20)–(23)). These constraints will be relaxed and their effect explored in a following article.

This hybrid individual/community strategy of managing resources is found in various parts of the world and is known to be frequently more cost-effective than when silvicultural practices are managed at the community level alone (*Kijima, Sakurai & Otsuka, 2000*; *Rana & Chhatre, 2017*). However, the model carefully considers the possibility to modulate and extend the forest governance and collective behavior of resource users in the PI (see Supplemental Data and Code). This way, the model can also be used to explore a variety of forest institutional agreements, econometric behaviors, and ecological contexts encountered across the planet. For instance, the equations in the model could easily be replaced to account for other institutional agreements, such as the community forest enterprises (CFE) found in Mexico (*Antinori & Rausser, 2007*), the mixed state-community forms of joint forest management (JFM) found in many developing countries (*e.g.*, *Baker, 1998*; *Sakurai et al., 2004*), the iriaichi-yamawari mixed community-private system of Japan (*Kijima, Sakurai & Otsuka, 2000*; *Shimada, 2014*), the mixed association-private agreement systems found in many European countries (*Korhonen, Hujala & Kurttila, 2012*), or the American cooperation model between non-industrial private neighbors (*e.g.*, *Klosowski et al., 2001*; *Vokoun et al., 2010*). All of them would ultimately provide a different range of PI values for the regulation and support parameters on the RI (equations from links 4b, 5b, *etc.*).

### Collective weeding and thinning model (arrow 4a in Fig. 1)

Weeding and thinning efforts are considered in the collective model as support activities that are undertaken by the association of RUs to free space and to provide light gaps, water and nutrients for the tree species of livelihood interest. The model does not consider invasive/alien species from outside the predefined mix of species, grass or animal species. The econometric model of thinning and weeding developed by *Sakurai et al. (2004)* was used since it is simple, replicable, and considers that weeding is not a separate effort on different target species but, instead, an extra collective effort consisting of thinning the

dominant tree species that is not of use for timber harvesting and the least interesting for fuelwood. The model also assumes a two-step decision process for thinning/weeding. It first considers that a global effort is allocated at the stand level, which depends on the governance system, the distance to the forest and the size of the forest (*Sakurai et al., 2004*: Decision 1). The model then considers that the association of RUs in the PI collectively allocate the weeding/thinning effort at the species level. This effort depends on the capabilities of the RUs and the time to collectively detect the most dominant species among others (*Garrard et al., 2013*: Decision 2). In this study, these variables are considered to be fixed and parameterized on the basis of the most common or average values found in the literature.

*Decision 2.4.1.1—Weeding and thinning effort at the forest stand level*
The collective weeding/thinning effort in the model depends on a series of factors (*Sakurai et al., 2004*; *Joshi & Arano, 2009*): the governance of forestry practices (PRIVATE: binary variable 0-1, so 0 in our case), the time it takes to walk to the forest ($T_{wt}$), the size of the forest parcel ($F_{area}$), and the presence of gravel ($S_{gravel}$: binay 0-1) and loam ($S_{loam}$: binay 0-1):

$$Thinning = 6.09 \pm 2.23\,PRIVATE - 2.34 \pm 1.2\,F_{area} - 4.37 \pm 1.79\,S_{gravel}$$
$$- 3.33 \pm 1.71\,S_{loamy} \sim 0.4 \qquad (20)$$

Then, as previously mentioned, an extra effort was applied to control the same invasive species that is not of any use for timber and fuelwood. The target of this weeding effort can be changed if an invasive species is identified and taken into account. Considering the distance to the forest, the simple weeding effort model of *Sakurai et al. (2004)* (man-day/ha/year) is defined as follows:

$$Weeding = 30.9 \pm 12.2 - 2.79 \pm 1.35\,log(T_{wt}) \qquad (21)$$

*Decision 2.4.1.2—weeding/thinning effort at the species level*
Once the man-day effort is defined, the association of RUs is faced with the problem of detecting and controlling the dominant species with a specific set of LHTs, among various other species with other LHTs (*Garrard et al., 2013*). The complexity of this multi-species/trait problem has a cost that translates into an increase in time to detect the dominant species (*Garrard et al., 2013*). The model of *Garrard et al. (2013)* gives a first approximation of the time of detection between plant species. However, this model should of course be updated once more general rules are known for trees. Assuming that all RUs are experienced observers (EXPERT: binary variable 0-1), and that the density of individuals per square/m is equal to the frequency of occurrence at one hectare, the model of *Garrard et al. (2013)* gives a good primer to define the time in minutes to detect one individual of the dominant species:

$$t_{mean} = e^{(4.27 \pm 0.58 - 0.17 \pm 0.39\,EXPERT - 0.28 \pm 0.12\,log(-log(1-N_{survey}) + 0.34 \pm 0.17 + 0.27 \pm 0.2 + 0.89 \pm 0.2))} \qquad (22)$$

Finally, to predict the actual number of trees detected and weeded/thinned ($N_{thinned}$), the model simply makes, like *Garrard et al. (2013)*, the reasonable assumption of a spatial

random distribution of the trees, and values of 8 h per man-day and 60 min/h (*i.e.*, t = 480 min), such that:

$$N_{\text{thinned}} = \frac{480(\text{Weeding} + \text{Thinning})}{t_{\text{mean}}} \qquad (23)$$

***Control of the access for individual harvesting and post-harvesting replanting to a diversity of resource users (links 5b and 5b' in Fig. 1)***

For this article, RUs possess the right to harvest timber and enough deadwood for individual household heating (fuelwood). Given that harvesting more timber can impact fuelwood availability and that fuelwood is critical for their livelihoods, RUs need to respect basic, collectively-defined rules concerning the access rate for timber harvesting (link 5b) and for fuelwood (link 5b'). The effect of changing links 5b and 5b' is symmetrical and explored in this article, which means that an RU who harvests live trees for timber can collect deadwood to heat his/her own household. On the other hand, the impact of link 5b' is defined as a fixed constraint in the model, where timber harvesting must stop if fuelwood harvesting cannot at least reach the minimal livelihood threshold (see Eq. (16)). However, as was mentioned before, this level was never reached in the simulations that we explored, and will be studied in a subsequent article.

## Model analysis and simplification

The main control variable of the CIS model is the access rate to the RI by more RUs. By increasing the access rate, the model assumes that opening the forest CPR to a greater coexistence of RUs produces a greater diversity of management disturbances on a greater proportion of competing tree species (through independent harvesting and post-harvesting planting on each species). This study explores the link with implied life-history trait (functional) diversity on the above-ground carbon biomass carbon stock, and on the collective economic return of timber supply from a greater diversity of species based on international market prices. Finally, the relationship is analyzed to emphasize the functional role of a reduced set of tree life-history traits.

As can be inferred from the modeling details presented above, the entire CIS model can be considered to be of intermediate complexity. Consequently, the numerical results obtained across a wide range of situations were quite long to produce. Given the high dimensionality of the problem, it was necessary to strategize the exploration. Five strategies were used:

(i) The first strategy was to proceed *ceteris paribus*, *i.e.*, by assuming some fixed social-ecological constraints (collective governance of the common-pool resources). Rather than fixing them in an *ad-hoc* fashion, we used a mixture of empirical forestry behavioral models found in the literature in order to provide a primer for a range of realistic values of parameters for sub-models in UI and PI (harvesting, replanting, weeding maintenance, governance). The goal of this article was, in no case, to test the precise verisimilitude of their modeling outcomes. Instead, by establishing plausible parameters of forestry management on the basis of selected models, it was possible to reduce the
dimension of the general problem and focus on the exploration of the opening of forest access on forest outcomes to then study its sensitivity in response to two ecological conditions (in our case, wetter *vs* drier forest CPR) and, finally, to analyze the underpinning ecological structure and processes that drive these outcomes.

(ii) The second strategy was to constrain the RI model to eight types of species, corresponding to all of the combinations between two values for each of the three LHTs (see "Model of resource infrastructure (RI) and its internal dynamics (link 0a in Fig. 1)".): SLA (2.85 and 20 m$^{-2}$.kg), SWD (400 and 1,000 kg.m$^{-3}$) and SS ($10^{-7}$ and $10^{-3}$ kg). With the eight types of tree species considered, the effect of the $2^8 = 256$ combinations of tree species management were explored over 100-year projections of the composition, structure and dynamics of the forest and its outcomes. To facilitate generalization, criticism and comparison with future studies, the gradient of increased access is presented as a percentage/proportion of species managed for biomass (timber) extraction. In the figure captions, the actual number of species is specified (*i.e.*, one species = 12.5% managed, two species = 25% managed, …, eight species = 100% managed).

(iii) The third strategy was to limit the analysis to a reduced set of forest outcomes calculated every year: the expected average forest-level values for the three LHTs, the Shannon index diversity of these tree LHTs (proxy of forest tree diversity), the above-ground carbon biomass (MgC/ha/y), and the revenue generated from timber harvested ($/ha/y).

(iv) The fourth strategy was to limit the sensitivity analysis to factors of interest. Following salient results from *Pichancourt et al. (2014)*, a similar sensitivity analysis was performed by focusing on the comparison of the impact of opening the access rates between two types of ecological factors: between wetter forest commons (that assumes water saturation in the soil bucket model from *Pichancourt et al. (2014)*), and drier forest commons (that assumes no water saturation and depends only on the rainfall pattern of the climate model from *Pichancourt et al. (2014)*).

(v) The fifth strategy was to focus on a deterministic model. Nevertheless, this article also provides the details of the behavioral Eqs. (1) to (23), the standard errors of their parameters, and the computer code to generate bootstrap confidence intervals, *e.g.*, 1,000 times, for specific scenarios (see Supplementary Data and Code). Computing them in a reasonable time for all the 512 scenarios (eight species, $2^8 = 256$ combinations of harvesting, times two types of forests: wetter and drier) would require proper code optimization associated with super computing capabilities and parallel programming.

## RESULTS

### Differential impact of opening the access to resource users at the forest-level outcomes between wetter and drier forest commons

Compared to non-harvested forests (in Fig. 3, "no access", equivalent to an exclusive conservation zone), opening the access to RUs who are disturbing species to supply timber and fuelwood in wetter forest commons is predicted to bring long-term economic benefits (Fig. 3A), stimulate regrowth and thus increase above-ground forest carbon biomass

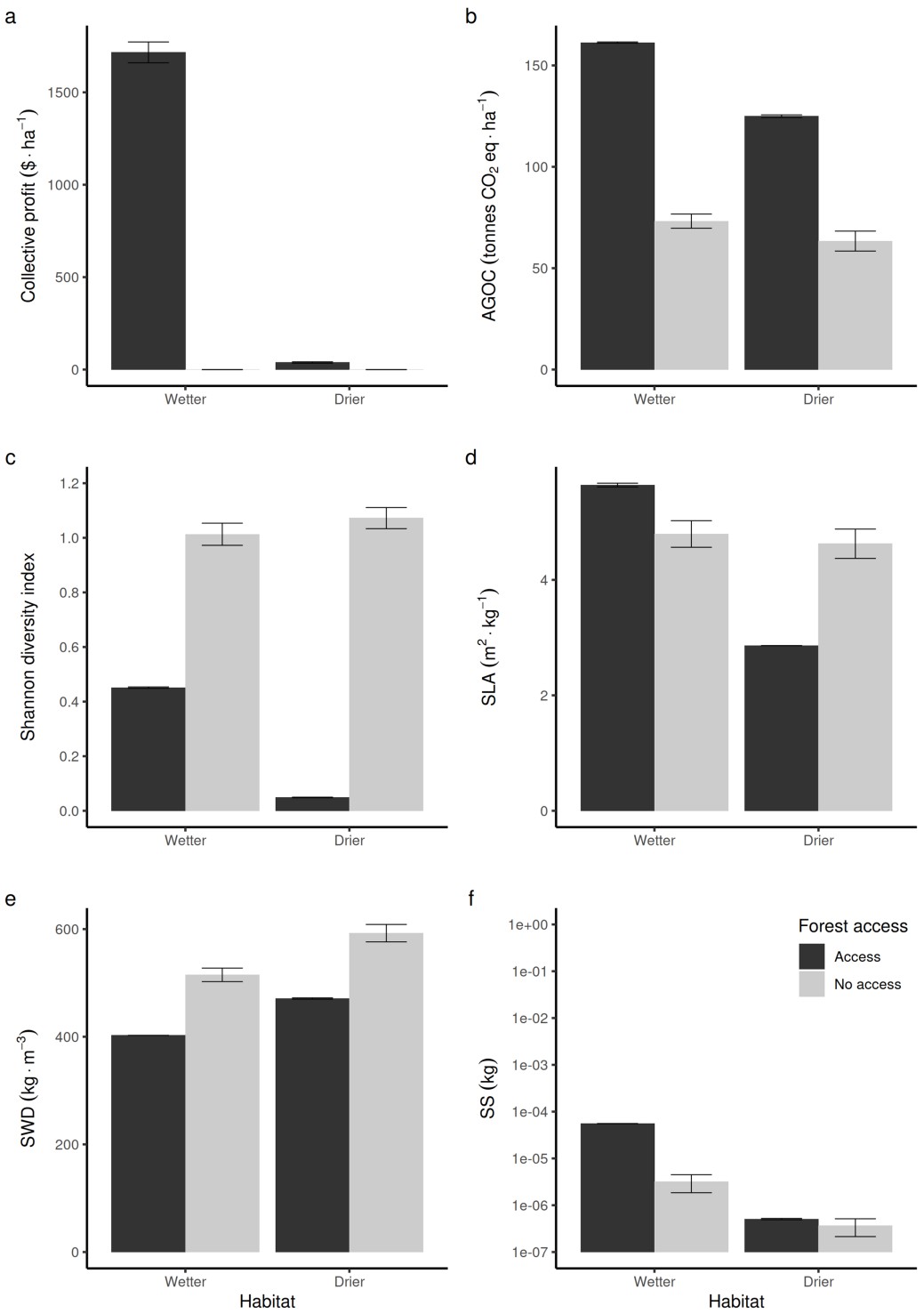

**Figure 3 Comparison of average forest-level outcomes between non-harvested and harvested forest commons in both (A, C and E) drier *vs* (B, D and F) wetter conditions: for (A and B) Shannon index of tree life-history trait diversity, (C and D) above ground organic carbon, and (E and F) collective profit from harvesting timber.**

(Fig. 3D). However, in that case, it reduces the long-term average Shannon index of biodiversity (Fig. 3C) by ~50%. In drier forest commons, the Shannon index of biological diversity is even predicted to be reduced by ~90%, without bringing much economic value.

In terms of life-history traits (LHTs), opening the access increases the average community-level SLA (Fig. 3D) in wetter forests, whereas it decreases on average in drier forests. The average specific wood density (SWD) is predicted to decrease both in wetter and drier forests when the access is opened (Fig. 3E). In contrast, the average community-level SS (and, thus, the average maximal size of tree species) produced in wetter forests is expected to increase, whereas it is not predicted to be affected in drier forest commons (Fig. 3F).

## Differential impact of increasing the access to a greater diversity of resource users on forest outcomes between wetter and drier forest commons

### Average and variance of the impact

Opening the access of the forest to a greater diversity of RUs, each specialized in harvesting and replanting a different species, is expected to impact the three main forest outcomes in both drier and wetter forest commons. The Shannon index of LHT diversity is expected to reach its maximum level at different proportions of species targeted by management disturbances in drier (Fig. 4A) and wetter (Fig. 4B) forest commons. In drier ones, this maximum level is generally low, highly variable and reached at intermediate levels (~37.5%), whereas in wetter ones, it is expected to reach a much greater maximum value and when most species are controlled by management disturbances.

Forest carbon storage is expected to be less impacted in drier forests (Fig. 4C) than in wetter forests (Fig. 4D), even though the variability of the results is greater in drier than in wetter ones. In wetter forests, increasing the proportion of species targeted by management disturbances always leads to greater long-term levels of living above-ground carbon storage than in non-harvested forests. Nevertheless it still leads to an expected reduction in carbon.

Profit generated from harvesting is obviously expected to increase with the number of species targeted by management disturbances for timber harvesting. However, profits are expected to be much greater in more productive wetter forests (Fig. 4F) than in drier ones (Fig. 4E).

Detailed impacts on the equilibrium dynamics between the three forest outcomes can be seen in Fig. S3 (S3.1 for drier forests and S3.2 for wetter ones).

## Understanding the sensitivity of the impact on the basis of the relationships between the average measures of life-history traits observed at the forest level

The impact was then broken down by emphasizing the role of the three main life-history traits measured at the forest level. This impact is highly predictable and is very sensitive to forest wetness. The results are broken down into three analytical steps.

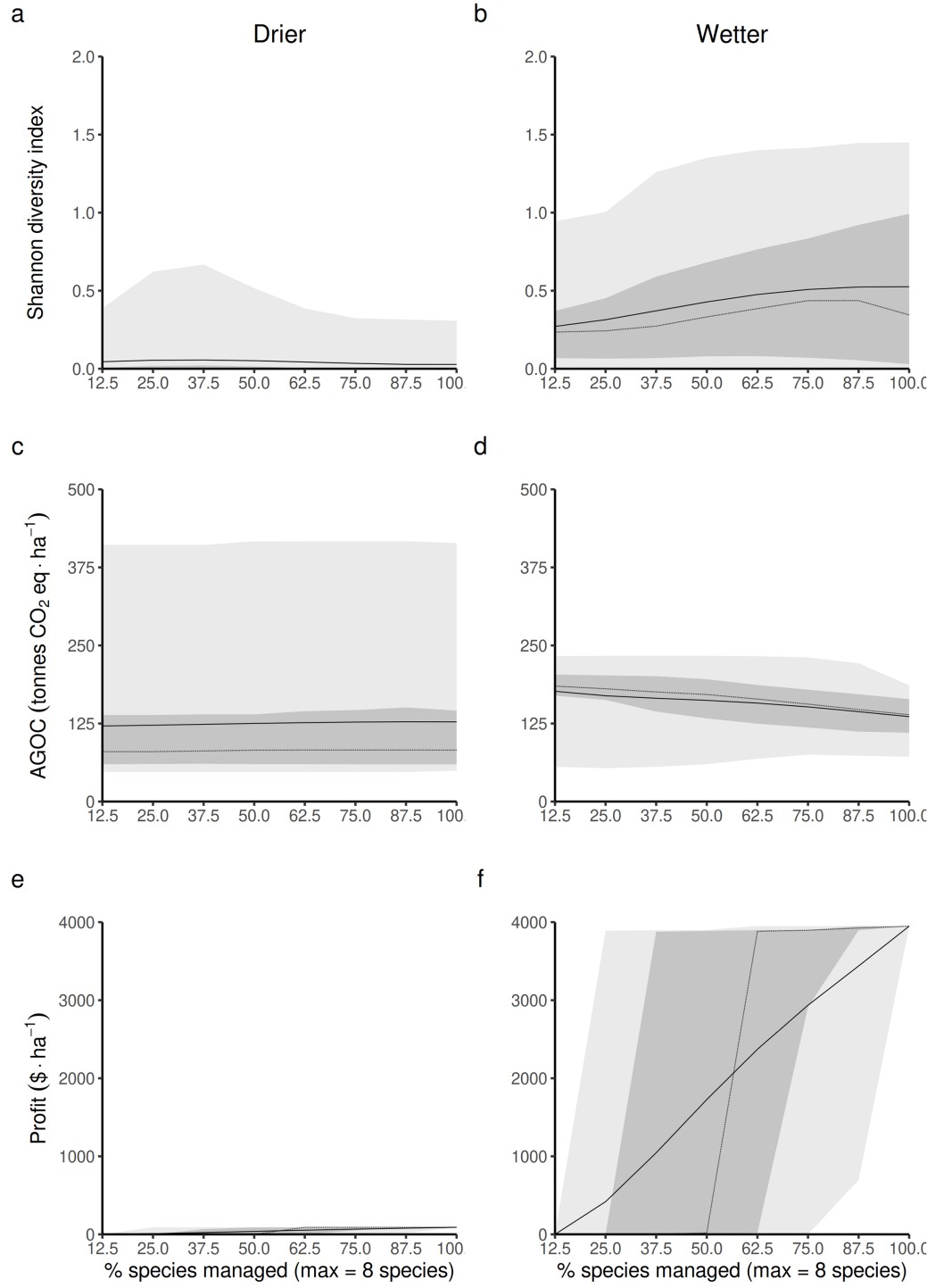

**Figure 4 Predicted relationship between changing the forest access to a greater diversification of species resource users harvesting and replanting a diversity of species, and three forest-level outcomes, in both drier (A, C and E) and in wetter (B, D and F) forest commons: (A and B) the long-term total revenue from timber harvesting (based on international timber market price), (C and D) the long-term average Shannon diversity of tree species (defined by their life-history traits), and (E and F) the long-term average above-ground organic carbon biomass (AGOC).** The solid and dotted lines represent the median and mean trend over 100 years of management regime. The

**Figure 4** (continued)
dark and light grey ribbons represent the quantiles [0.025 0.25 0.5 0.75 0.975], *i.e.*, the interquartile range and 95% confidence interval. Here the proportion of species harvested is based on a forest containing eight species, *i.e.*, one species = 12.5% managed, two species = 25% managed, …, eight species = 100% managed.           

### *Impact of opening to a greater diversity of species resource users on the average value of forest-level life-history traits (LHTs)*

The relationship was first broken down to emphasize the impact of changing the access strategy on the dynamics of the three life-history traits. The model predictions were sensitive to whether it was in drier or wetter forests.

In drier forest commons (Figs. 5A, 5C and 5D), only SWD is predicted to be sensitive to changing the percentage of species managed for timber harvesting (Fig. 5C), where the maximum value of SWD is predicted to be obtained for intermediate proportions of species disturbed (~1/3 species harvested, *i.e.*, between two-three of the eight species selected).

In wetter forest commons (Figs. 5B, 5D and 5F), SWD is predicted to only be slightly sensitive to this change (Fig. 5D). Both SLA and SS are predicted to be reduced (Fig. 5B) and increased, respectively (Fig. 5F) in a symmetric fashion by increasing the proportion of species managed for timber harvesting: the maximum SLA and minimum SS values are predicted to be obtained at a minimum proportion of species managed, whereas the minimum SLA and maximum SS values are predicted to be obtained at a maximum proportion of species managed.

### *Structural constraints between average forest-level life-history traits (LHTs)*

The second step was to focus on how the species assemblages in the forest commons are predicted to evolve in terms of the relationship between the average LHTs. When plotting all the years, scenarios and forest wetness classes, we can detect the same non-linear trade-offs, mostly visible between SLA and SS, and between SLA and SWD (Fig. 6). In fact, the model predicts that forests with greater average SLAs should be composed of trees that also produce smaller average seeds and lighter timber (and *vice versa*). A breakdown of drier and wetter forests shows that the latter should be sufficient to detect these above-mentioned trade-offs between LHTs.

Figure S4 (S4.1 in drier and S4.2 in wetter) provides details on how the difference between LHT trade-offs dynamically change between scenarios broken down and averaged per class of proportion of species managed (from 0% to 100% of the eight species from the forest). For instance, in wetter forest commons, opening the access to RUs and then increasing the proportion of species managed is predicted to change the LHT pattern. When there is no access, the model predicts that the domination by shorter trees producing smaller seeds should dynamically alternate between trees with different SLAs and SWDs along the dynamic trade-off in an unpredictable way (Fig. S4.2: 0%). Then, when opening the access to RUs (Fig. S4.2: >0% and <100%), the model predicts that wetter forest commons should have more complex dynamics between high and low values for the three LHTs, but progressively with greater SS (taller tree species) since the access to

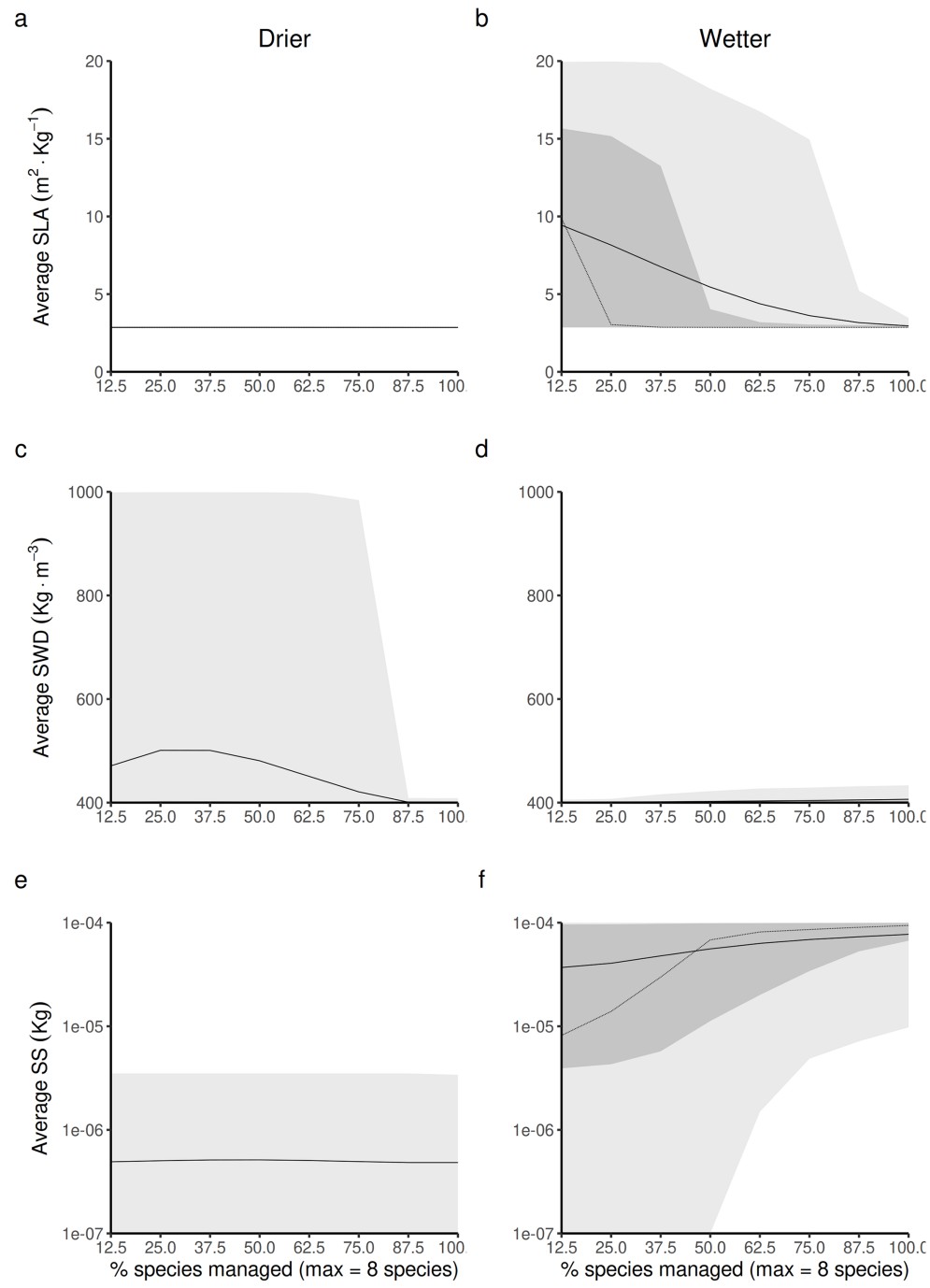

**Figure 5 Expected relationship between changing the forest access to a greater diversification of species resource users harvesting and replanting a diversity of species, and three life-history traits indexes measured at a forest-level, in both drier (A, C and E) and in wetter (B, D and F) forest commons: (A and B) the specific leaf area (SLA) index, (C and D) the specific wood density index (SWD), and (E and F) the seed size index (SS).** The solid and dotted lines represent the median and mean trend over 100 years of management regime. The dark and light grey ribbons represent the quantiles [0.025 0.25 0.5 0.75 0.975], *i.e.*, the interquartile range and 95% confidence interval. Here the proportion of species harvested is based on a forest containing eight species, *i.e.*, one species = 12.5% managed, two species = 25% managed, …, eight species = 100% managed.

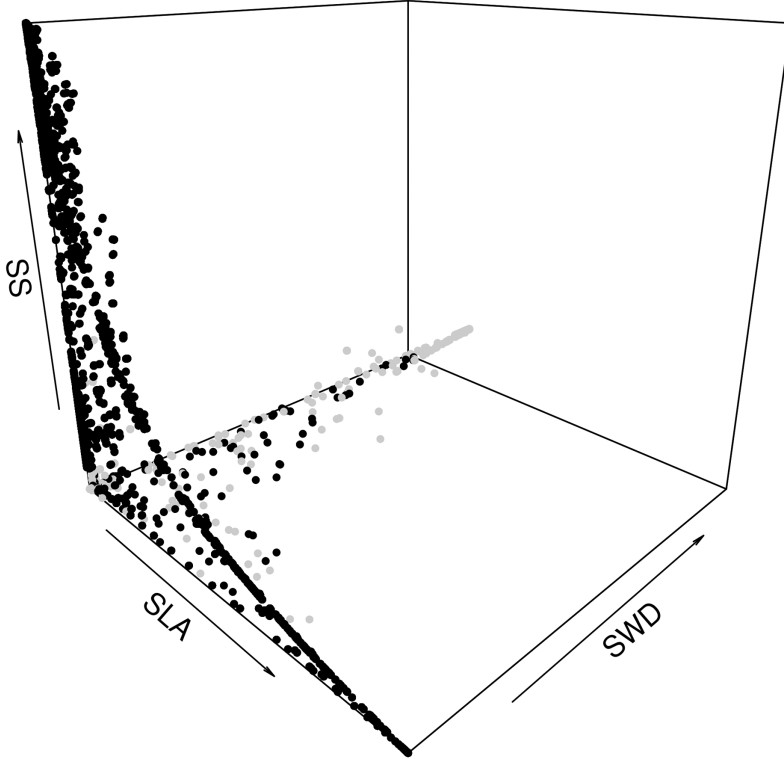

**Figure 6 Expected relationship between three average forest-level indexes of life-history traits, in both drier (grey points) and wetter (black points) forest commons: the specific leaf are index (SLA), the specific wood density index (SWD), and the seed size index (SS).**

a greater diversity of RUs increases. Finally, when all species are managed for timber extraction (Fig. S4.2: 100%), the tree species generally dynamically and predictably alternate between different values of SLA and SWD (with stable high SS values), in such a way that the complete dynamics gravitate around a stable attractor point (SLA ~ $m^2.kg^{-1}$, and SWD ~ $kg.m^{-3}$; see Fig. S5) and form a limit-cycle. More details on the temporal trajectories of the individual LHTs can be found in Fig. S5 (S5.1–S5.6).

### Impact of these life-history constraints on biological diversity, carbon stock and timber profit

The second step was to analyze how forest biodiversity, carbon and economic outcomes from timber production were affected by the change in forest-level LHTs. The effect on biodiversity is detailed below (Fig. 7). The effects on carbon and economic outcomes are presented in Figs. S6.1 and S6.2, respectively. It was possible to predict three laws that linked the Shannon index to the three LHTs. The shape of these laws does not change with wetness. Only the position along this law changes with wetness.

When looking at the average forest-level SLA, the Shannon index is predicted to reach its maximal value at intermediate to low average forest-level values of SLA (~5 $m^2.kg^{-1}$) by following a logistic law (Figs. 7A and 7B). The turning point of the maximal Shannon index (Fig. 7B) corresponded in Fig. 5B (wetter forest commons) to ~2/3 of the species

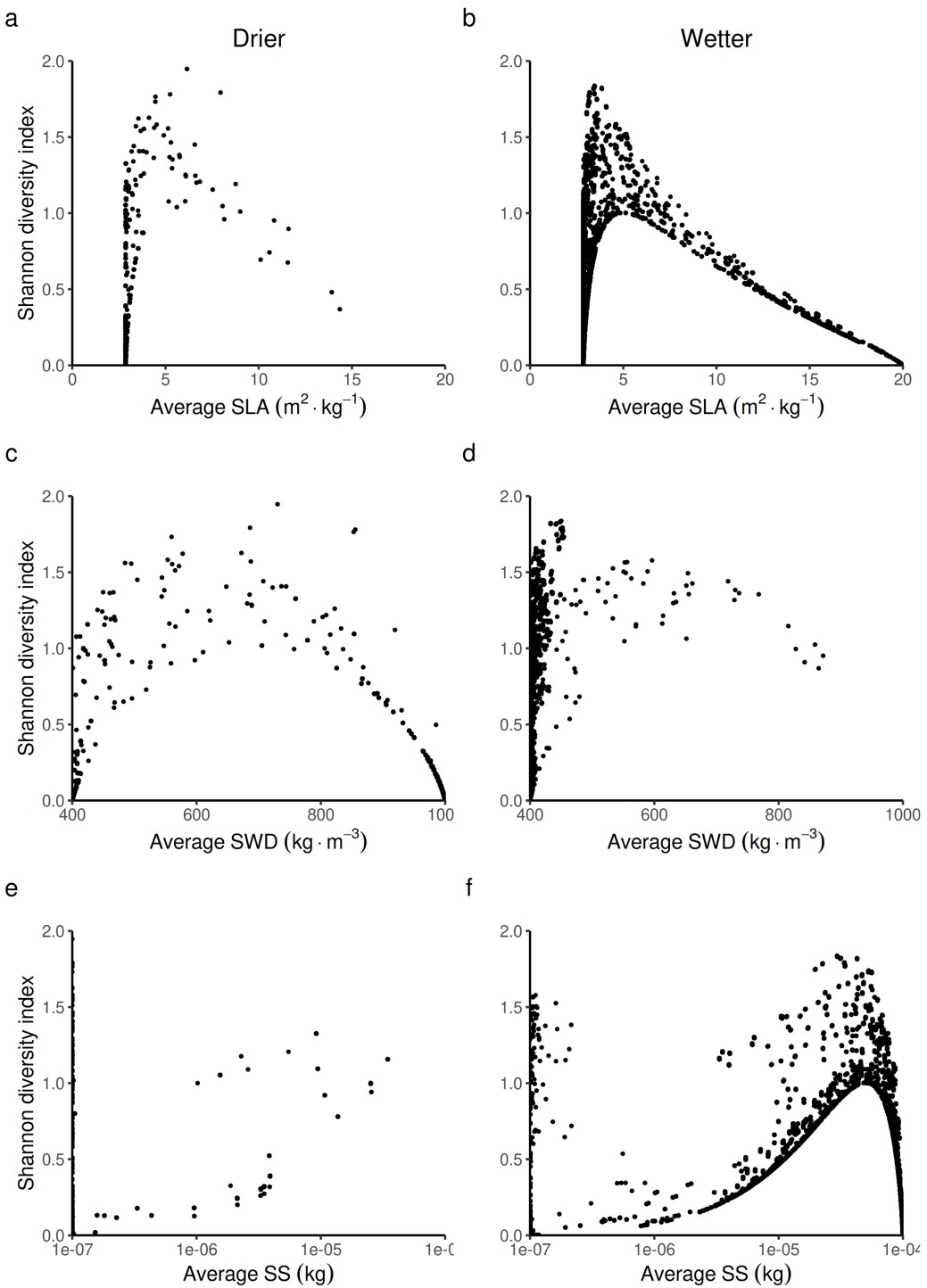

**Figure 7 Expected relationship between the Shannon index of biological diversity and three average forest-level indexes of life history traits, in both drier (A, C and E) and in wetter (B, D and F) forest commons: (A and B) the specific leaf area (SLA) index, (C and D) the specific wood density index (SWD), and (E and F) the seed size index (SS).**

managed. The entire spectrum of values in that case homogeneously covers the law from low to high average forest-level values of SLA. Therefore, the turning point is expected to be detected in these wetter systems. In contrast, in drier forest commons (Fig. 7A), the lower Shannon index observed in Fig. 4 can now be explained by the fact that average forest-level SLA values rarely reach (and even less often exceed) the SLA turning point of the maximal Shannon index, regardless of the percentage of species managed for timber harvesting.

When looking at the average forest-level SWD, the maximum Shannon index in drier forest commons is expected to be reached roughly halfway along the SWD spectrum. It follows a parabolic law (~700 kg.m$^{-3}$, Fig 7C), even though most values fall more frequently into either high or low SWD states, rarely sustaining in-between SWD states. For this reason, the turning point of the maximal Shannon index (~700 kg.m$^{-3}$, Fig. 7C) is rarely reached and is reduced to a narrow range of percentages of species managed for timber harvesting (~1/3 [17.5–40%] of the eight species managed; see Fig. 5C). In comparison, wetter forest commons (undisturbed and disturbed ones) follow a similar law, but with, in addition, much lower SWD values that cover any type of Shannon index values (actually corresponding to the varying effect of SLA and SS laws).

Finally, when looking at the average forest-level SS, the Shannon diversity index in drier and wetter forest commons once again follows the same logistics law. However, compared to the SLA law, the maximal level of the Shannon index is reached for forests with high average SS values (> 5.10$^{-5}$ kg). This level is only reached in wetter forest commons (Fig. 7F), generally when ~2/3 of species are managed for timber harvesting (see Fig. 5F), even though any proportion of species managed (between 12.5% to 100%) can regularly achieve this result. In contrast, drier forest commons mostly contain tree species that produce small seeds (10$^{-7}$ kg: Fig. 7E), which can be reached for any proportion of species harvested for timber (Fig. 5E). Moreover, averagely low SS forest commons that are only composed of small tree species that produce small seeds (10$^{-7}$ kg) can produce any Shannon index values from among the possible ones (Figs. 7E and 7F), corresponding to the effect of the SLA law (Figs. 7A and 7B) and the SWD law (Figs. 7C and 7D).

## DISCUSSION

It is challenging to find a proper theoretical framework to help find laws and understand their causal origin regarding the sustainable coexistence between biodiverse carbon-rich ecosystems and socially diverse human communities that need to equitably and profitably access these common-pool (bio-)resources (*Ostrom, 1990*; *Poteete & Ostrom, 2004*; *Naidu, 2009*; *Saunders, 2014*; *Betts et al., 2021*). The goal of this article was to contribute to the solution of this social-ecological problem using a mathematical model of multi-species forest dynamics.

It is demonstrated here that when such a model is constructed around the CIS framework, it can be used to strategically reframe the general social-ecological problem into a new ecological hypothesis that is far easier to study (see Introduction). By re-framing the problem this way, it was possible to perform pure ecological analyses, and to then interpret the general social-ecological problem in relation to classic ecological theories.

## Conjectural empirical laws linking the diversification of resource users to the level of biodiversity

The CIS-based model of multi-species forest dynamics used here first predicted that forests opened to targeted human-controlled disturbances can be biodiverse, but always less than in protected forests, regardless of the conditions (Fig. 3). A more significant result was that increasing the diversity level of (human-controlled) disturbances should increase the Shannon index of tree LHT diversity, the above-ground carbon stock and expected economics benefits in wetter tropical forests, as opposed to forests where only one of the species is harvested by one user (and this, regardless of the species). Sustainable management could be achieved, but outcomes were sensitive to the moisture conditions of the forest common.

These results need to be compared to a large empirical dataset from a wide range of existing social-ecological contexts, which is obviously hard to obtain. Nevertheless, they already corroborate various observations from forestry science. For instance, greater species and functional diversities (especially in the understory) are observed when forests are subject to repeated selective logging disturbances, or other disturbances that are frequent enough to prevent competitive exclusion over an entire area, but not frequent enough to eliminate most species (*e.g.*, *Berry et al., 2008*; *Calbi et al., 2021*). Similarly, forestry scientists have shown that applying a diversity of controlled disturbances on mixed species stands (known as *irregular shelterwood*) is now known to adapt canopy openings to the requirements and tolerances of a greater diversity of tree species, *e.g.*, shade-tolerant ones (*Daniel, Helms & Baker, 1979*), and without sacrificing vigorous small merchantable stems (*Raymond et al., 2009*). Irregular shelterwood is usually studied as a coordinated strategy by a single forestry actor (*Berry et al., 2008*; *Raymond et al., 2009*), and sometimes in relation to a mixture of completely unrelated random natural drivers (*e.g.*, fire, wind, diseases; *Calbi et al., 2021*). However, a non-centralized vision (like in our case) can equally be suggested where a greater diversity of species can emerge and coexist with disturbance practices of a diversity of non-coordinated RUs working in the same forest common. The reason for expecting these patterns to coexist is that in human groups, no single member can possess the entire spectrum of knowledge, beliefs and practices (KBP) to sustainably harvest and manage a highly biodiverse CPR that covers different ecological conditions by himself (*Poteete & Ostrom, 2004*). Instead, human communities are characterized by a diversity of coexisting actors that possess different socio-economic and cultural attributes in relation to the independent extraction of different biological resources (*Natcher, Davis & Hickey, 2005*; *Lemos & Agrawal, 2006*), in order to equitably secure a diversity of forest ecosystem benefits (*e.g. Agrawal & Gibson, 1999*; *Toledo et al., 2003*; *Kimengsi et al., 2019*). The efficacy of these institutional agreements for sustaining social-ecological outcomes have been well-documented empirically (*e.g.*, *Persha, Agrawal & Chhatre, 2011*). However, theoretical modeling studies are still needed to explore coexistence situations beyond what has been observed. The present model and results offer this opportunity to predict the existence of such a scientific law.

We now address the issue of how the results can be explained using existing ecological theories. In the following sections, we demonstrate that three ecological theories can successively be used: the life-history theory (summarized in *Caswell, 2000*), the forest succession theory (*Lohbeck et al., 2013*; *Poorter et al., 2019*), and Chesson's species coexistence and disturbance theory (*Chesson, 2000*; *Barabás, D'Andrea & Stump, 2018*). To use them efficiently to interpret the results, the best way is to start by not separating (*i.e.*, accept to conflate) the effect of disturbance regimes in our results. Their interpretation in "Structural constraints and causal mechanisms on species co-existence" will then serve as a baseline for explaining in "Structural constraints and causal mechanisms on social-ecological co-existence" what is happening when we instead focus on the impact of the diversification of disturbance regimes.

## Structural constraints and causal mechanisms on species co-existence

In the example of wetter tropical rainforests, tree species compete only for light and respond to climate variabilities. In that case, life-history theory predicts that tree species with unique combinations of LHT values (SWD, SLA, SS) possess unique life cycles and demography to respond to vertical light gradients (*Caswell, 2000*). Therefore, few tree species are expected to have exactly the same vertical niche space (see detailed explanation in *Pichancourt et al. (2014)*). On the basis of this simple principle, combined with Chesson's species coexistence principles, it is clear that the greater coexistence and vertical partitioning of species along the light gradient should preferably be achieved through "*stabilizing mechanisms*" (*Chesson, 2000*; *Barabás, D'Andrea & Stump, 2018*). Translated into the LHT/functional dimension, this mechanism predicts that along the vertical light niche space, species coexistence should be greater if the competition pressure between species with the same SLA is greater than between species with different SLA values (and independently of their differences in SWD and SS values).

This process was observed in the wetter forest results where the variation of light-promoting traits like SLA (and SS) play a major role in niche partitioning and positive species coexistence (Figs. 5–7). In particular, the model predicted that the Shannon diversity index was greater and more variable in forests that were dominated by tall trees (*i.e.*, producing seeds with greater SS; *Pichancourt et al., 2014*) and leaves with lower SLA (for irradiation tolerance) (Figs. 7B and 7F). The simple reason is that having a dominant upper story is not incompatible with having an understory that contains a diversity of tree species characterized, on the one hand, by a diversity of lower SS seed traits that cover a wider spectrum of small tree heights, and on the other hand, with a diversity of greater SLA leaf traits that cover a wider spectrum of shade tolerance.

Using the same *stabilizing mechanistic* explanation, we can explain predicted differences in the level of species coexistence between wetter and drier forest commons, where the latter type of forests are expected to promote the coexistence of more species with greater and more variable SWD values (Figs. 5C, 5D, 6C, 6D, 7C and 7D). This pattern of changes in the SWD is symptomatic of the wet/drought tolerance dimension, and corroborates previous findings as well (*Poorter et al., 2019*).

The same reasoning can also be extended to interpret temporal results along the forest dynamical succession of these traits. Indeed, the model predicts that the dynamics of community-level trait values (Fig. S5) are also constrained by strong trade-off laws (Fig. 6) that drive species coexistence/diversity (Figs. 7A and 7B). For instance, in wetter environments, as forests are transitioning to secondary successional stages (with taller species producing bigger seeds and lower SLA), the coexistence of species is expected to increase as well (Fig. S4.2), corroborating previous findings (*Lohbeck et al., 2013*; *Muscarella et al., 2017*). More relevantly, standard successional theory also predicts that successions would also proceed from low towards high forest-level SWDs in wetter forests (and on the contrary, from high towards lower community-level SWDs in drier forests), once again following the law that high SWD well reflects drought tolerance in harsh early successional environments (*Poorter et al., 2019*). This was actually corroborated by our independent theoretical predictions in drier (Fig. S5.3A) and in wetter forest commons (Fig. S5.4A).

## Structural constraints and causal mechanisms on social-ecological co-existence

The mechanistic explanation differs when we analyze the effect of the diversification of harvesting and replanting disturbances on forest outcomes, even though the structural constraints maintain their relevance here.

Generally speaking, indiscriminately opening a forest creates light gaps. In this case, niche partitioning disappears. Consequently, the dynamic coexistence between species will then depend more on the relative competitive abilities, *i.e.*, the relative population growth rate of each species to quickly harness light (SLA), in order to avoid being trapped in a non-viable vertical niche space. In the model, this competitive ability depends on the traits that affect all life stages of the plant (Fig. 2): the seed traits that promote fast germination, the cheapest wood traits that benefit wood construction to resist drought, and the cheapest leaf traits that boost growth under strong light to quickly reach maturity and produce seeds earlier. In our case, the most competitive species that is expected to win this race is the one with the lowest SLA (2.85 $m^2.kg^{-1}$), lightest SWD (400 $kg.m^{-3}$), and lowest SS ($10^{-7}$ kg) (*i.e.*, shortest trees producing many small seeds).

This dominant species occupies a relatively greater basal area than the other species. Consequently, by harvesting it when the minimal forest biomass threshold is reached (see "Decision 2.3.1.1—stand level preference: what is the probability that the plot is logged?" of Methods) means opening the relatively largest light gap, thus freeing the relatively greatest quantity of resources for other species to grow (space, light and soil water). As a consequence, more species can coexist with different LHT combinations, especially in wetter forests where only light traits define relative dominance (*Pichancourt et al., 2014*). This process of increasing species coexistence by controlling dominance corresponds to the second "*equalizing mechanism*" described by Chesson in his theory (*Chesson, 2000*; *Barabás, D'Andrea & Stump, 2018*). The fact that drier forests would not necessarily lead to greater species coexistence is due to the selection pressure that provides opportunities for greater SWD species to resist successional drought earlier, increase both inter- and

intra-LHT competition, and thus lower SS and SLA tree species (due to the LHT trade-offs; see Fig. 6).

Opening the access to a greater number of RUs specialized in different species increases the probability that the most abundant and competitive tree species is controlled. By following this probabilistic argument, it is easy to understand that for every new RU added to the forest CPR, there is an increased probability that another new dominant species is harvested and replanted. Successive additions further promote the *equalization* benefits, giving opportunities for more species to compete on equal ground (Fig. 5). Because the species LHT dominance can change with environmental context (water, light) and successional stage, the benefits are expected to be maximal when forest access is opened to as many and diverse RUs as there are species to harvest and replant (obviously under the collective governance constraints on management practices specified in the Methods section). Conversely, if the forest is opened to homogeneous RUs that target only one species (and especially the less dominant one), then biodiversity is expected to drop drastically. The tree assemblage will then be predominantly dominated by the relatively few dominant ones (with greater SS and lower SLA and SWD for wetter forests, and lower SS and SLA and greater SWD in drier forests). This is what is predicted in Fig. 3 when only one species is randomly selected to be harvested and replanted.

Altogether, we can see that the relationship between some forms of biodiversity and livelihood diversification processes can simply be explained using human-controlled mechanisms that have their theoretical equivalent in three ecological theories: the disturbance, succession, and Chesson coexistence theories. However, the effect of these mechanisms and their success at achieving good social and ecological outcomes are fundamentally structured by existing constraints between LHTs.

## Extending findings and methodology to solve related CPR dilemmas

Even though underpinning invariants can be predicted, their impact on the success of finding positive outcomes was sensitive to the moisture conditions. Other ecological factors may thus justify other sensitivity analyses, *e.g.*, in response to changes in climate, soil structure, maximum number of species, *etc*.

Changing human management and governance rules (links in Fig. 1) are also known to impact carbon and biodiversity outcomes in forests. However, how does relaxing these factors affect these outcomes (Figs. 3 and 4; Fig. S3) and their predictions on the basis of their relationship with the structural constraints (*e.g.*, Figs. 5–7 and Figs. S4–S6)? A set of factors are of particular interest and their study may benefit from being framed by the newly discovered set of constraints and causal mechanisms. For instance, the assumption of *Ostrom (1990)* of perfect collective monitoring (CPR principle 4) and enforcement (CPR principle 5) of the CPR access against free-riders (PI, links 4, 5 and 6 in Fig. 1) should be relaxed, as their validity is debatable, like in many tropical countries (*e.g.*, *Agrawal & Gibson, 1999*; *Chhatre & Agrawal, 2008*; *Finer et al., 2014*; *Brancalion et al., 2018*).

Likewise, the model used what could be called a *diversified disturbances strategy* (hypothesis). However, comparing it to other disturbance strategies may also help to better frame long, ongoing debates associated with the disturbance theory (*e.g.*, *Chesson, 2000*;

*Fox, 2013*; *Barabás, D'Andrea & Stump, 2018*). For instance, how does it compare to a maximalist strategy where RUs are collectively forced to eradicate the same most dominant species (see, *e.g.*, *Chesson, 2000*; *Pichancourt et al., 2014*; *Barabás, D'Andrea & Stump, 2018*), compared to the reverse conservationist strategy where only the most threatened ones cannot be disturbed or to another strategy where the intensity, frequency or area of disturbances (harvesting and/or replanting) are optimally calibrated at intermediate levels?

Additionally, how does grouping all RU activities into the same CPR location like is done here (a strategy called *land sharing*) compare to another strategy where activities are separated into different spatial zones (a strategy called *land sparing*)? The promotion of one or the other is based on fundamentally different beliefs on how human disturbances are impacting species coexistence and productivity, suggesting that the present method may also help to solve this similarly ongoing debate (*Fischer et al., 2014*; *Runting et al., 2019*).

Finally, the approach could be used to understand the state of social and ecological diversity in areas that have been nurtured by culturally-diverse groups. For instance it can be used to evaluate whether the process of diversification of disturbance practices can explain the state of local biodiversity (*e.g.*, *Heckenberger et al., 2007*; *McKey et al., 2010*; *Maezumi et al., 2018*), and reversely the state of local social diversity (*e.g.*, *Loring, 2016*). By synthesizing these symmetric problems, the approach could open up interesting avenues for the science of "biocultural diversity" when exploring the inextricable links between the processes of biological (genes, phenotypes, species) and cultural (knowledge, beliefs and practices) diversification (*Maffi, 2005*).

### Funding
This work was supported by the Institut National de Recherche pour l'Agriculture, l'Alimentation et l'Environnement (INRAE). The funders had no role in study design, data collection and analysis, decision to publish, or preparation of the manuscript.

### Grant Disclosures
The following grant information was disclosed by the authors:
Institut National de Recherche pour l'Agriculture, l'Alimentation et l'Environnement (INRAE).

### Competing Interests
The author declares that they have no competing interests.

### Author Contributions
- Jean-Baptiste Pichancourt conceived and designed the experiments, performed the experiments, analyzed the data, prepared figures and/or tables, authored or reviewed drafts of the article, developed the mathematical model and code, and approved the final draft.

## Data Availability

The code for the model and data for Figure S2 are available in the Supplemental Files.

## Supplemental Information

Supplemental information for this article can be found online at http://dx.doi.org/10.7717/peerj.14731#supplemental-information.

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
