# Peer review of "Some fundamental elements for studying social-ecological co-existence in forest common pool resources"

_PeerJ, doi:10.7717/peerj.14731_

## Round 0.1 · original submission · Major Revisions

The present study presents the development of a socio-ecological model for forest resource use by forest dwellers, based on an existing forest dynamics model. Two reviewers assessed the manuscript, and are positive (and so am I) but have a number of comments that will need to be addressed before the manuscript can be accepted. I would like to particularly emphasize (i) clarifying the effect of the assumptions, (ii) assessment of uncertainty as recommended by reviewer 1 and (iii) clarifying the text (incl. language) and have it double checked, ideally by someone else, before resubmission.

I add a few comments of my own below:

The biocultural part falls a bit short of the expectations, and my recommendation would be to either strengthen or reduce that part (e.g. in the introduction).

The monosectoral perspective on livelihood diversification may be severely limited for real communities where farming and fishing are also important (e.g. Ellis & Allison 2004), and increasingly other activities (commercial, tourism etc). This could be discussed.

L31 unclear which debate is being referred to. If important, summarize in a sentence.
L42 the numbers might have changed quite a lot since 2004? (also Word->world)
L47 examples of species?
L51 references for successes and failures?
- In Fig. S3 it is not clear which points are mahogany and which not.

·

Basic reporting

Clarity:
- The paper is nicely written but there are still many spelling and/or grammatical errors: some examples below:
L.18 played a role
L. 24 like
L. 37 consequence
L.44 despite … being
L. 63 engaging in
L. 82 designed for
I thus suggest careful proofreading.
- I would suggest re-writing some long sentences to make them easier to understand and more straight-forward. For instance L. 90: “I particularly suggest the non-trivial conjectural statement that one of the causal process that increases biological diversity in forests may be through the process of diversification of independently-controlled disturbances regimes on more species, imposed for livelihood outcomes » could be changed to “In particular, I test the hypothesis that forest biological diversity is increased by livelihood diversification through a higher number of harvested species in controlled disturbances.”
- I would also suggest using more consistent terminology throughout the text (e.g. give the same sub-model names in the text L. 126-137 and figure 1).

Figures:
- Figure legends should be more explicit and explain all elements of the figure. For instance, in Figure 1, the legend doesn’t explain what “species-level”, “plot-level”, etc refer to.
- In my opinion, Figure 1 is essential to help the reader understand the model, but could be better organized to have key elements easily visible and with elements such as model inputs, how elements of submodels interact, etc. All these should appear in different colors or with specific arrows, etc. to make the organization of the diagram very clear to the reader, and so that information is easy to find.
- You could potentially add an additional layer of information in Figure 2 by coloring data points according to the total number of species (from 0 to 8) harvested in each of them.
- Figure 3 should be provided with a higher quality (the current one is relatively pixelated and some error bars are difficult to see). I would also suggest adding a color legend (scenarios) in the figure (in addition to the description in the legend), for example in the empty bottom half of the last panel.
- Figure 3: Specify clearly that all 8 species types are being ‘disturbed’ (less ambiguous than ‘a higher diversity of species’ in the legend).

Literature:
- Some citations are not accurate: for example, L. 76, there is no mention in Runting et al. 2019 to any diversification of livelihoods (the paper considers protected areas, selective logging, and plantations with no mention to the number of species or local communities).
- Some references are missing from the reference section (for example Runting et al. 2020, cited twice)
- L. 430: add citations for the LHT and the DT.

Background:
I would suggest adding to the introduction (L. 94-98) a more comprehensive background on the model used, and on other models of forest dynamics and/or socio-economic models, and why this one was chosen– comparative strengths, processes included, main assumptions, etc. I think this is particularly important as the main findings of the paper are all based on this model.

Experimental design

The paper is an interesting modeling approach to the question of socio-ecosystems sustainability in the context of tropical forests. The model presented in this study ingeniously mixes socioeconomic modeling with forest ecology, a multidisciplinary approach that is much needed to answer some of the most crucial questions in forest management. This approach offers some new insights into the debates over harvest diversification and land sharing vs. land sparing in tropical forests and shows that it is theoretically possible (under some specific conditions) to maintain high levels of tree species diversity and carbon while allowing a diversity of harvests for timber and fuelwood.
I enjoyed reading the paper, but my main concerns are twofold: (i) the clarity of the paper could be much improved (and in particular the methods section, which is essential to interpret the results); (ii) there is no estimation of uncertainty nor of the sensitivity of the results to some strong assumptions, many of which are unrealistic in most tropical regions.
Model codes are provided (although I didn’t see them cited in the main text) making the methods reproducible. However, the methods section in the main text lacks some key information (mentioned below) and is generally difficult to follow. I strongly suggest reorganizing this section, adding missing information, and potentially adding a figure or two with additional diagrams describing submodels 1 and 2 (diversification and behavioral models).

There is no assessment of uncertainty in the study; I suggest estimating the distribution of parameters (or their level of uncertainty) whenever possible and doing some error propagation through the entire model, for example with the Monte Carlo method.

Information missing in the methods:
1. Please provide a description of the 8 types of species harvested (L. 312), or cite Figure 4 (it took some time for me to find the information).
2. Description of scenarios used in Figure 3 (L. 340-354) should go in the methods.
3. L. 316-318: what model are the climate predictions based on?
4. Model parameters should appear more clearly in the main text and/or in figure 1. In particular, parameters that vary from one simulation to the other should be apparent.
5. It is not clear how harvests (for timber or fuelwood) affect the rest of the forest in the model, and in particular logging damage, infrastructure building, etc are not mentioned.

Validity of the findings

Some strong assumptions were made in this study (in particular L. 177-182), among which the legality of timber harvests and the strong governance, the validity of which is debatable in many tropical countries (e.g. Brancalion et al. 2018 doi: 10.1126/sciadv.aat1192; Finer et al. 2014, doi: 10.1038/srep04719). This limitation is mentioned in the discussion, and I understand it is not the main point of the paper to explore this subject, but given that these assumptions probably determine the findings of the paper, I think it is important to make some of these assumptions more apparent (I would suggest mentioning them as soon as the introduction). Additionally, I would suggest testing the sensitivity of the model to those assumptions by modifying the parameters associated with them (e.g. lower probability of engaging in silvicultural practices under weaker governance and land tenure) and quantifying the changes in final results (this could be provided in the SI).

Reviewer 2 ·

Basic reporting

The “Abstract” title is missing and please make in one graph The line numbers also need to be started in the abstract section.
Introduction: Line 2? Please use formal and consistent space between texts or paragraphs.
Lines 22 and 32: Which one is correct, bioculture or bio-culture? Please use the correct term consistently. Furthermore, what mean by biological diversity, biocultural diversity, livelihood diversity, species diversity, linguistic diversity and cultural diversity their relations? In some cases, one is the subset of other. Please be specific to avoid confusions in the first place.
The English language should be improved to ensure that an international audience can clearly understand your text. Some examples where the language could be improved include lines 11, 20, 386, 412, the phrase “inextricably linked”. Additionally, in some parts I suggest to use passive construction that sounds impersonal tone and more formal than the use of active voice (such as ‘I’).
Line 82: Replace “traded off” with “traded-off”.
Line 84: The phrase “Triple bottom line” needs clarity.
Figure 2 is not clear and Figure 4 is not well visible.
Line 362 delete “species”.
Lines 509-518: Check the font type/size.

Experimental design

no comment

Validity of the findings

no comment

Additional comments

The method and discussion sections are too long.

---

## Round 0.2 · Minor Revisions

I'm taking over this article from the previous editor. I'm aware it was reviewed previously, and the submitted version has done a good job addressing many of the previous reviewers comments. As I was not able to have previous reviewers comment on the paper, I invited an additional reviewer. Their comments are below -- there is less of a concern with methods, and more with writing / clarity. The majority of these comments are focused on the introduction/discussion, but the whole paper could use another round of editing for clarity (perhaps also asking a colleague for a friendly review/edit would be beneficial). I think the methods and some of the discussion are more clear because sections are bulleted out with headers.

·

Basic reporting

I think the study itself should be worthy of publication at some point, but there is a need for more work to present the work in a clear coherent and succinct manner. This reads like a very wordy and often unclear draft. The language is generally correct but there is a lack of focus and clarity that is both a problem in itself and also an obstacle to offering clearer comments -- I am not always sure I grasp the ideas.
I cannot invest time in rewriting but have indicated a few examples and comments on the text (see PDF).
In general terms the text can be reduced by a third or more and most sentences can be made clearer and sharper.
Overall I was frustrated: this was difficult to read and not ready to submit ... even though it is already a second round. Should it have been sent to review? I think not. I skimmed after the first few pages. Asking for feedback from trial readers and editorial assistance may be valuable.

Experimental design

Unsure. As above greater clarity would permit me to offer more ...

Validity of the findings

Unsure. As above the clarity and focus needs to be greatly improved. I did not dwell on this.

Additional comments

As I already noted above I found this article frustrating as it was wordy and often opaque. Most of us write and rewrite many times before we submit something. Reviewers should not be asked to give detailed feedback on something that was not ready to submit ... it is not a correct use of our time and reduces our willingness to help in future. If language is an issue then asking feedback from trial readers and editorial assistance may be needed.

---

## Round 0.3 · accepted · Accept

Thanks for taking the time to shorten the paper significantly, and edit the paper for clarity. This new version is much improved.